# Asking physicians how best to implement cervical cancer prevention services in India: A qualitative study from Mysore

Prajakta Adsul[1,2]*, Sasha Herbst de Cortina[1,3,4], Rashmi Pramathesh[1], Poornima Jayakrishna[1], Vijaya Srinivas[1], Suzanne Tanya Nethan[5], Kavitha Dhanasekaran[5], Roopa Hariprasad[5], Purnima Madhivanan[1,6,7,8]

**1** Public Health Research Institute of India, Mysore, India, **2** Division of Epidemiology, Biostatistics, and Preventive Medicine, Department of Internal Medicine University of New Mexico Comprehensive Cancer Center, Albuquerque, New Mexico, United States of America, **3** School of Public Health, University of California Berkeley, Berkeley, California, United States of America, **4** School of Medicine, University of California Irvine, Irvine, California, United States of America, **5** Division of Clinical Oncology, ICMR-National Institute of Cancer Prevention and Research (NICPR), Noida, India, **6** Health Promotion Sciences Department, Zuckerman College of Public Health, University of Arizona, Tucson, Arizona, United States of America, **7** Division of Infectious Diseases, Department of Medicine, College of Medicine, University of Arizona, Tucson, Arizona, United States of America, **8** Department of Family and Community Medicine, College of Medicine, University of Arizona, Tucson, Arizona, United States of America

* PAdsul@salud.unm.edu

**Data Availability Statement:** As a qualitative manuscript, all data are submitted as a part of the article.

## Abstract

Cervical cancer is the second most common cancer among Indian women. Screening is an effective prevention strategy, but achieving high screening rates depend upon identifying barriers at multiple levels of healthcare delivery. There is limited research on understanding the perspectives of providers who deliver cancer prevention services. The objective of this study was to explore physician perspectives on cervical cancer prevention, barriers to effective implementation, and strategies to overcome these barriers in India. Guided by the "Multilevel influences on the Cancer Care Continuum" theoretical framework, we conducted semi-structured interviews with physicians in Mysore, India. From November 2015- January 2016, we interviewed 15 (50.0%) primary care physicians, seven (23.3%) obstetrician/gynecologists, six (20.0%) oncologists, and two (6.7%) pathologists. We analyzed interview transcripts in Dedoose using a grounded theory approach. Approximately two-thirds (n = 19, 63.3%) of the participants worked in the public sector. Only seven (23.3%) physicians provided cervical cancer screening, none of them primary care physicians. Physicians discussed the need for community-level, culturally-tailored education to improve health literacy and reduce stigma surrounding cancer and gynecologic health. They described limited organizational capacity in the public sector to provide cancer prevention services, and emphasized the need for further training before they could perform cervical cancer screening. Physicians recommend an integrated strategy for cervical cancer prevention at multiple levels of uptake and delivery with specific efforts focused on culturally-tailored stigma-reducing education, community-level approaches utilizing India's community health workers, and providing physician training and continuing education in cancer prevention.

**Funding:** Research reported in this publication was supported by the Fogarty International Center of the National Institutes of Health [under training grant # TW009338 to PA]. The content is solely the responsibility of the authors and does not necessarily represent the official views of the National Institutes of Health. The funders had no role in study design, data collection and analysis, decision to publish, or preparation of the manuscript.

**Competing interests:** The authors have declared that no competing interests exist.

## Introduction

Cervical cancer is the second most common cancer among women in India.[1] According to the World Health Organization 2018 estimates, approximately 96,922 new cases and 60,078 deaths were due to cervical cancer in India, accounting for nearly one fifth of the global cervical cancer deaths [1]. While the widespread uptake of Human Papillomavirus (HPV) vaccination is important for cervical cancer prevention, screening remains an important parallel strategy to reduce the cancer burden, especially in low- and middle-income countries (LMICs) such as India. The impact of screening is dependent upon achieving high screening coverage as well as ensuring that screen-positive women receive appropriate follow-up care. High screening rates, in turn, are influenced by factors at multiple socio-ecological levels as outlined in the National Cancer Institute's Multilevel Model, which proposes that national, state, and local policies; healthcare organizations, providers, and teams; and individuals in the family and community can influence cancer care delivery [2]. Identifying and understanding the influence of these multilevel factors can guide the development and testing of interventions for effective, safe, timely, efficient, equitable, and patient-centered healthcare delivery.

Understanding multilevel influences requires input from multiple stakeholders and the use of multiple research methods to uncover the complexity of healthcare delivery. According to India's Fourth National Family Health Survey in 2015–2016, less than 16% of women aged 30–49 in the Mysore District reported ever being screened for cervical cancer [3]. Several studies have conducted qualitative research to understand patient perspectives on utilizing cervical cancer screening services in India and other LMICs [4,5]. A majority of these studies reveal significant barriers at the levels of the provider or healthcare system. For example, a study conducted in Mangalore, Karnataka, reported that women who saw their physicians in the past five years did not receive education regarding cervical cancer and screening from their physicians [6]. Physicians, especially those working in primary care, play an important role in promoting the uptake of cancer prevention services, with previous research suggesting a doctor's recommendation to be a strong predictor of whether or not a woman engages in cancer screening [7–9]. Therefore, targeting delivery of screening requires a comprehensive understanding of physicians' roles and interactions with their staff, perspectives about individuals and communities seeking care, and experiences with the organizations in which they work.

However, there is limited research in the context of the Indian healthcare system and perceptions of physicians regarding cancer screening services. Prior studies have taken a quantitative approach to evaluate physicians' knowledge, attitudes, and practices; and found gaps in provider education, training, and resources [10–12]. Investigating physicians' perspectives using qualitative methods can build upon this works to identify and address the factors that influence implementation of cervical cancer prevention programs in their communities. Given this background, our aim was to understand perspectives of a group of physicians (including primary care physicians, obstetrician/gynecologists, oncologists, and pathologists) that could be involved in providing cervical cancer related preventive services in Mysore, India.

## Material and methods

### Theoretical background

The "Multilevel influences on the Cancer Care Continuum" framework provided the theoretical background for this study [13]. The interview questions were developed in line with this model and were kept as broad as possible to query about physicians' perceptions on implementing clinical preventive services for cervical cancer in their respective healthcare organizations and their communities. Where appropriate, we probed about factors on multiple

socioecological levels (individuals, communities, healthcare organizations) and asked about strategies that could overcome barriers for implementation in their practice settings. Using a qualitative approach allowed researchers to generate an in-depth understanding and enhanced our ability to gather detailed information from physicians with busy schedules. The interview guide is available in Supplementary Material.

## Setting

The study took place in Mysore, India, a district in the state of Karnataka, between October 2015-January 2016. As of the Indian National Census in 2011, the district has a population of 3,001,127, and approximately 58.0% of residents live in rural villages [14]. In India, approximately half of the population receives healthcare in the private sector, and the other half through the public healthcare system [15]. Within the public healthcare system, the central government sets national standards and provides funding for programs to be implemented by states, through district-level efforts. India's Operational Framework for the Management of Common Cancers, published in 2016, recommends visual inspection with acetic acid (VIA) every 5 years for women aged 30–65 years, to be performed at Primary Health Centers by Medical Officers (primary care physicians), nurses, and mid-level providers including Auxiliary Nurse Midwives (ANMs) [16]. The Public Health Research Institute of India (PHRII), which conducted the current study, implemented community-based mobile cervical cancer screening using VIA, reaching over 3,000 women between 2010–2015. However, at the time of interviews, there was no organized district-level cervical cancer screening program. To our knowledge, there is no literature on the scope of cancer screening practices among healthcare providers prior to this study.

## Participant recruitment

PHRII and Florida International University Institutional Ethics Boards reviewed and provided approvals before the start of the study. We utilized a convenience sampling approach by recruiting physicians who had existing relationships with PHRII, followed by snowball sampling, with the first set of physicians helping recruit additional participants. All individuals contacted for the study were practicing physicians in Mysore district and gave written informed consented to be interviewed. We approached potential participants by phone and in-person. We included physicians from both private and public sectors and a variety of specialties (i.e. primary care physicians, obstetricians and gynecologists, pathologists, and oncologists) to incorporate diverse perspective from providers that could be involved in delivering cervical cancer screening for the women in the community. Our goal was to capture both aspects of the screening process—delivery of the tests as well as the results—and therefore, we included pathologists in our study. After each interview, the interviewers de-briefed and recorded field notes from each interview. Based on this debriefing and field notes, we stopped recruitment when we noted no additional concepts being discussed by the participants (i.e. theoretical data saturation, which indicates that the analysts discovered no new concepts in their iterative review of transcripts [17]).

## Data collection and analysis

The authors PA and RP, conducted the interviews in English and took field notes; both are female and introduced themselves as affiliated with PHRII. Most participants were aware of PHRII's work in community-based cervical cancer screening. Interviews were conducted in-person, at a time and location convenient to the participants, most often in their clinical offices, and lasted approximately 20–30 minutes. All interviews were audio-recorded and transcribed verbatim

before analysis. PA and RP checked all transcripts for accuracy without returning them to participants for comment or correction. Transcripts were analyzed using Dedoose, a web-based software that can be used to code, organize, and analyze qualitative data [18]. Analyses followed a grounded theory approach, which is a well-accepted methodology to study social phenomena and follows an open coding process to discover emerging concepts as reported by the participants [19]. To further clarify our application of this approach, we detail the steps undertaken for this analysis. Two analysts (PA and SH) individually reviewed all transcripts to gain an overall understanding of the data and generate a preliminary list of codes that reflected important concepts as discussed by the participants. We used this preliminary list to code four transcripts, meeting on a regular basis to discuss, combine, and add new codes as needed. We agreed upon a final list of codes that SH applied to the remainder of the transcripts. PA reviewed 20% of the transcripts (n = 6) at random to ensure codes were applied consistently. As coders, we achieved consensus through meetings to discuss coded excerpts and resolve discrepancies in coding. We reviewed the excerpts for codes and summarized trends and relationships for each code, while distilling the major themes from the data and present them in the results.

## Results

### Participants

As shown in Table 1, 15 primary care physicians, seven gynecologists, two pathologists, and six oncologists participated in the study. All primary care physicians worked in the public sector,

**Table 1. Characteristics of thirty physicians in Mysore, India interviewed regarding strategies for cervical cancer prevention (2015–2016).**

| Variables | Primary care physicians (n = 15) | Obstetricians and Gynecologists (n = 7) | Oncologists (n = 6) | Pathologists (n = 2) |
|---|---|---|---|---|
| Median age (range) | 42 (34–52 y) | 44 (34–66 y) | 39 (33–46 y) | 52 (50–54 y) |
| Gender, No. | | | | |
| Male | 10 | 0 | 2 | 1 |
| Female | 5 | 7 | 4 | 1 |
| Median years of practice (range) | 11 (1–24 y) | 13 (3–35 y) | 8 (2–20 y) | 22 (18–26 y) |
| Type of org, No. | | | | |
| Public hospital/center | 15 | 0 | 2 | 2 |
| Private hospital/clinic | 0 | 7 | 4 | 0 |
| New cases per day, No. | | | | |
| Less than 10 cases | 0 | 0 | 3 | 1 |
| 10–20 cases | 2 | 5 | 3 | 0 |
| 20–50 cases | 8 | 1 | 0 | 0 |
| More than 50 cases | 5 | 1 | 0 | 1 |
| Cervical cancer patients per month, No. | | | | |
| Less than 10 | 15 | 7 | 4 | 2 |
| More than 10 | 0 | 0 | 2 | 0 |
| Currently opportunistically screening, No. | | | | |
| Yes | 0 | 4 | 1 | 2 |
| No | 15 | 3 | 5 | 0 |
| CME attendance, No. | | | | |
| Last three years | 7 | 6 | 6 | 2 |
| 3–6 years | 3 | 1 | 0 | 0 |
| More than 6 years ago | 5 | 0 | 0 | 0 |

and none reported providing any cervical cancer screening. Only oncologists reported routinely caring for cervical cancer patients, with a minority seeing over ten per month.

The analysis revealed five themes, described below.

**Theme 1. Promoting an understanding of preventive care and health maintenance among women is necessary to overcome barriers to cervical cancer screening.** Several physicians described low awareness around cancer prevention that affected utilization of preventive services, specifically for gynecologic health. Study participants commonly reported women did not understand the female reproductive organs, specifically considering that the local language *Kannada* does not have a direct translation for the word "cervix," as described by this physician–

> "Translating in *Kannada* language, I mean we don't have any translation, like we say '*garbhakosha*'; garbhakosha is actually uterus."–Participant #1 (Oncologist working in the public sector)

Physicians reported that the concept of screening for cancer was not familiar to many of their patients even though cancer diagnoses were common in the community. Women usually associated visiting a doctor, only when experiencing symptoms like pain or disfigurement that interfered with their daily activities, not with routine asymptomatic care, as described by this primary care physician–

> "Unless it affects their daily life or when symptoms reach to the peak level, then only they will come with complaints. Because if we talk about breast, some nodules or fibroids, even about that they don't disclose anywhere (. . .) if we conduct a [cervical cancer screening] camp I doubt woman will come, I doubt."–Participant #9 (Primary care physician working in the public sector)

Another reason why women did not seek care according to participants, was the low prioritization of health maintenance for women. An Ob/Gyn described women in India as "silent sufferers" as follows–

> ". . .women are silent sufferers in India. . .They are never given the care which they are due for. Even if she is suffering post coital bleeding or white discharge, foul smelling discharge, [she] won't be taken to primary healthcare center at all."–Participant #4 (Ob/Gyn working in the private sector)

In some cases, participants noted that women did not have the autonomy to seek healthcare services since most were financially dependent on their husbands and needed permission from husbands or mothers-in-law. Physicians explained that women prioritized household duties and care for other family members, largely influenced by the cultural upbringing in India, as described below–

> "[If] the husband or somebody at home or the children, they fall sick, [the women] just look into their problem. They just see that they get well as early as possible. But same thing is [happening] to them, they just neglect it."–Participant #5 (Ob/Gyn working in the private sector)

Additionally, the costs of coming to the clinic were often too high–not only the direct cost of healthcare, but also finding childcare, taking time from their household responsibilities, paying for transportation, and lost wages for both women and other family members to accompany

them to the clinic. The physicians concluded that improving prioritization of women's health maintenance would be necessary to achieve high screening coverage, as noted below–

"Actually, it is very difficult to convince them because they have to come all the way, living there, one day work and it will be loss for them and most of the people will refuse."–Participant #16 (Primary care physician working in the public sector)

**Theme 2. Reducing stigma around gynecologic health and improving trust in and the value of cancer services is necessary for increased screening uptake.** Participants reported a specific unwillingness among women to seek care for gynecologic issues. They commonly described women feeling shy or embarrassed to discuss gynecologic symptoms and refusing to undergo pelvic exams even when they had signs of disease, as described by this primary care physician–

"Sometimes they will be having pain [in the] abdomen or white discharge or bleeding. . .so because of shyness they will not come to hospital, because of shyness they will not come to hospital."–Participant #11 (Primary care physician working in the public sector)

Furthermore, physicians stated that women sometimes worried about negative perceptions from their families and communities regarding gynecologic health problems. They reported peers to be key influencers for women seeking care or participating in screening programs. This could be a positive influence, such as when women came to the clinic after being urged by friends, or when more women became more comfortable discussing and receiving gynecologic care after their peers had positive experiences. However, some participants cautioned that screening programs would not work if women shared negative or misleading experiences with their peers, as may be the case notes by an oncologist–

"The message will spread fast; 'we don't go there, once you go there you will get discharge.' You know it is, we can't convince them, but the patients can convince the other patients or the other ladies. So they will not, they believe them and will not come for screening."–Participant #1 (Oncologist working in the public sector)

Other common misconceptions noted were pelvic examinations causing fainting, reproductive tract infections, or pain, which discouraged participation in screening programs requiring pelvic exams.

Many physicians described the negative perceptions about cancer that were prevalent in the community and could undermine screening efforts. They noted some instances where after receiving a cancer diagnosis, some of their patients were abandoned by their families, removed from household responsibilities, and even cases where they were divorced or separated from their husbands, as noted below–

"Once a wife [is] diagnosed [with] cancer, the husband would have left her and would have gone, because thinking that she is of no use. Once it is detected, cancer, he will just run away."–Participant #19 (Oncologist working in the private sector)

Participants felt that the possibility of getting a cancer diagnosis in fact discouraged many women from coming to screening, although some physicians stated that screening could be attractive to women when portrayed as a tool to prevent cancer and avoid the negative repercussions associated with a cancer diagnosis.

**Theme 3. Utilizing culturally-tailored community-level strategies to promote a culture of prevention and support the healthcare system in delivering cancer screening services.** The vast majority of participants highlighted the need for awareness and education regarding cervical cancer and screening. Given the context of low health literacy, gender roles, and stigma, physicians emphasized the importance of culturally-tailored strategies. Some physicians proposed reaching out through women's social groups, as noted below -

"They may have some festival, they meet each other or there may be kitty parties, housewives and all they may meet each other once a week or once in 15, whatever. So I feel instead of just having food and discussing about what sari you have taken, what dress you are taken, they can think about their health also. (. . .) Collaborate with some doctor, they will be knowing some doctor, they can always invite them and ask for some checkup"–Participant #7 (Pathologist working in the private sector)

Others suggested cervical cancer survivors could provide peer education for women in their communities, as described by an oncologist below–

"Cancer survivors, they can relate, they can tell the problem of not going to the doctor properly, they can tell that [the] way they have gone through [life] a disease has come. So by your preventive measure you can be a normal person. You may not have to [go] through what I have gone through"–Participant #2 (Oncologist working in the public sector)

A commonly recommended strategy was peer-led, community-based, educational programs that could also be delivered by or include female medical providers. Male physicians stated they could not implement screening programs unless female staff were present when performing pelvic exams. Multiple participants, especially male participants, outlined the role of female providers in creating a comfortable environment for pelvic exams, but required additional support for training staff who were often relied on in rural settings, as noted below–

"[It comes] down to female doctors, especially in this kind of symptom (. . .) in a rural setup that stigma is still there, the male and the female (. . .) if a staff nurse is there at least we will tell her 'you just examine and let us know,' and we do not know how much of knowledge she has regarding that."–Participant #15 (Primary care physician working in the public sector)

Another recurring theme was the role of community health workers (CHWs)–including Accredited Social Health Activists (ASHAs), ANMs, and *Anganwadi* workers, who work in nationally funded community-based centers for maternal and child health. Physicians noted that CHWs could provide education and counseling with broad community acceptance since they are usually from the communities they serve. Furthermore, because ASHAs, ANMs, and *Anganwadi* workers are predominantly female, they often have a better rapport and ability to discuss sensitive matters with other women and would be ideally suited to lead educational and motivational interventions and to track and encourage follow-up for patients who screen positive, as noted below–

"Awareness in the [community] that is the thing. . .we should train our ASHAs also, they are the ones who do house-to-house visits and then they live in the [community] only. . .and the Anganwadi workers, ASHA workers, ANMs. . .they will have the faith of the

people. . .if they tell them, then they will do it."–Participant #14 (Primary care physician working in the public sector)

**Theme 4. Physicians in the public sector described limited organizational capacity to provide preventive care in a system geared towards several public health concerns.** Many primary care physicians working in the public sector reported the lack of a prevention culture in their communities but also described limitations in their own capacity and their healthcare settings in providing clinical preventive services. They described a high workload, administrative and managerial duties, staff vacancies, and several ongoing public health programs, all of which they said would limit their ability in implementing screening services, as described by this primary care physician–

"As a Medical Officer our main role is to prevent diseases. . .we say 98% of our work is prevention and only 2% is diagnoses and treatment, but it has reversed actually so most of the time we will be sitting in the outpatient department (. . .) Mainly we have to concentrate on National Programs, we have to supervise National Programs. Apart from that we cover *Anganwadi* Health Checkups, (. . .) and antenatal, postnatal care and regular outpatient procedures and mainly National Programs."–Participant #13 (Primary care physician working in the public sector)

The high workload was not unique to physicians and nurses; CHWs in their settings were often similarly overcommitted and overworked. The time constraints and lack of resources cemented the need for physicians to focus on symptomatic care, limiting their ability to adopt new screening initiatives for asymptomatic women.

As physicians discussed the lack of time and resources available for screening programs, they commonly reached the same conclusion–implementation of cervical cancer screening would require substantial support in terms of funding, resources, and broader community collaborations. A common comparison was with the programs for tuberculosis control and polio vaccination which integrates community education, incentivized follow up from community health workers, and treatment from physicians themselves, as noted below–

"Just like polio [vaccinations]. . . Something like that because now polio is eradicated, so it was, it was like. . . they used to go to the house. Even after polio immunization they used to go to each house and ask whether the child was immunized or not. . ."–Participant #1 (Oncologist working in the public sector)

**Theme 5. Few physicians understood the basis of cervical cancer screenings, and most perceived a need for additional trainings.** Few obstetricians/gynecologists mentioned providing opportunistic screening when women came in for appointments, while describing specific issues with the Pap test reports. For example, a Ob/Gyn working in private sector mentioned,

"Prevention is main, screening has to be done, but I don't know while doing the screening [*which*] Pap smear reports are false positive or false negative. . .We get lot of [*reports*] indicating chronic inflammatory smears. . .so we don't know how to trust everything like, off course the specificity and sensitivity will be there with each and every test."–Participant #28 (Ob/Gyn working in the private sector)

The issues with Pap smears were also supported by the pathologists in the study, who described the need for well-trained physicians or staff in collecting Pap smears, as follows–

"So you need people who are trained, it could be nursing staff ideally the doctor should do it, but the doctor is very busy, he or she cannot waste time on a Pap smear, so he delegates the duty to the nurse, the senior nurse doesn't do it the trainee nurse they do it, so . . .I would say on many occasions the smears are not representative. . .because person doing it does not know where to take it [the sample] from."–Participant #18 (Pathologist working in the private sector)

None of the primary care physicians working in the public sector, however, reported screening their patients for cervical cancer. When asked about screening programs or efforts, some physicians discussed testing their patients for cervical cancer in the context of gyneco-logic symptoms. For example, a primary care physician reported -

"Many patients don't tell their signs and symptoms of [*cervical cancer*]. . .they are not telling it to all the doctors, they will keep inside. That is also important, patient has to tell first."–Participant #22 (Primary care physician working in the public sector)

Some participants acknowledged this practice was different from the recommended screen-ing guidelines, but others did not differentiate between screening and diagnostic use of pap smears, demonstrating a lack of familiarity with the principles of screening.

Many physicians stated that implementing cervical cancer screening would be appropriate, feasible, and acceptable in their practice settings. However, they noted that a major barrier to providing these services was the lack of skills to perform screening exams. Participants said they would need additional training before they would be able to collect samples for pap smears or perform visual inspection using acetic acid (VIA)/ Lugol's Iodine (VILI). For exam-ple, a primary care physician noted–

"But if I get a training. (. . .) Without [training] I cannot do that, even our staff will not do that. Here for this [IUD] insertion and all we are getting a pelvic periodic training and all. If you fellows give us a training then we will do all that".–Participant #24 (Primary care physi-cian working in the public sector)

Some primary care physicians noted that they often send women with gynecologic symp-toms to specialists because they lacked the necessary knowledge and skills related to gyneco-logical health. They stated that if they had a more consistent and higher volume of examinations–such as in a screening program–they would feel more confident to handle those cases themselves.

## Discussion

Physicians identified several factors influencing the implementation of clinical preventive ser-vices for cervical cancer. Based on the study findings, we created a visual display to depict the concepts and relationships emerging from the data in Fig 1. These insights from physicians focus primarily on how to increase the uptake of screening by individuals and the ability of providers to deliver screening services. Participants reported that women's willingness to undergo screening was embedded within the community context of their peer network, gender roles, and social norms regarding gynecologic health and cancer. Meanwhile, providers' ability to perform screening was influenced by the larger system of medical education, programmatic expectations, and human and resource capacity. Visualized in Fig 1, the factors highlighted at the multiple socio-ecological levels are reflective of the "Multilevel influences on the Cancer

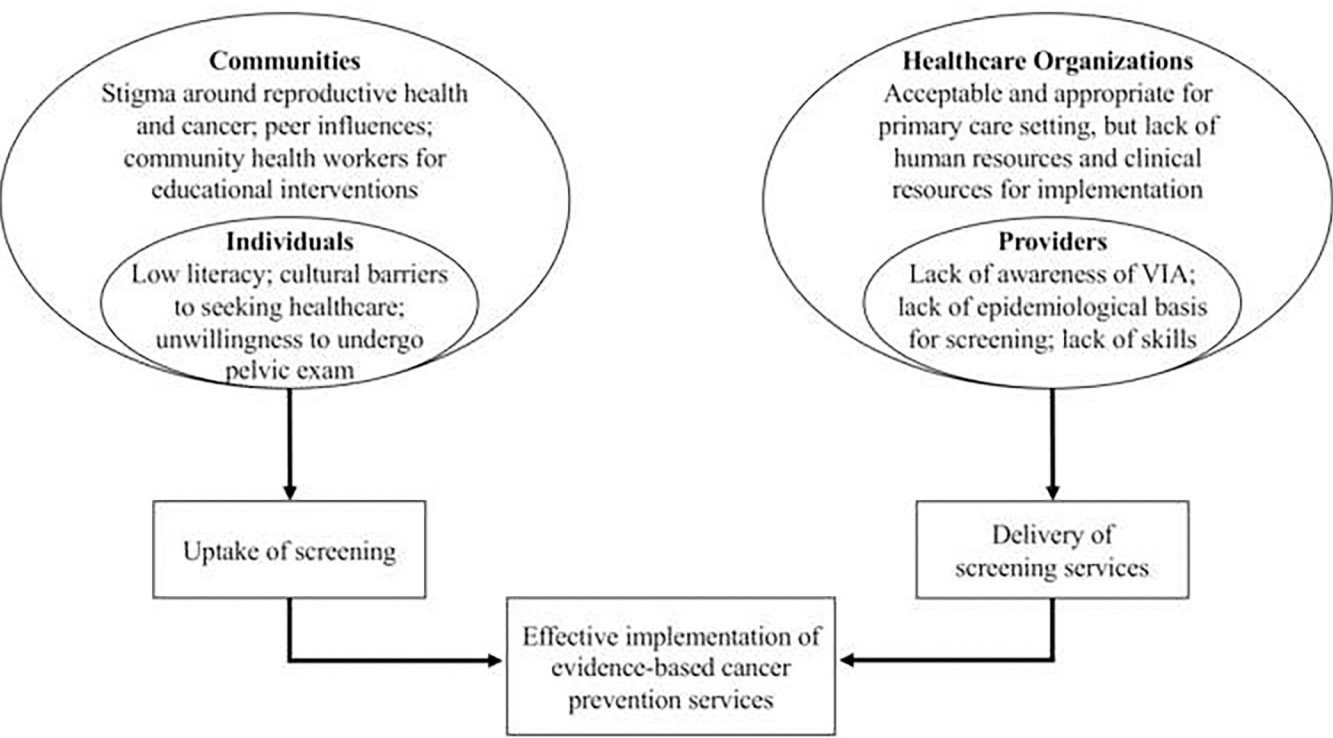

**Fig 1. Visual representation of concepts regarding implementation of cervical cancer prevention programs based upon interviews with thirty physicians in Mysore, India.**

Care Continuum" framework, which similarly underscores the embedded nature of cancer care delivery.

Physicians emphasized the need for coordinated implementation strategies in order to achieve high screening coverage within the cultural context and the healthcare system. Simply addressing screening through isolated programs targeting, for example, provider rates of offering screening or women's knowledge regarding asymptomatic diseases, would not suffice. Rather, the physicians described a comprehensive approach that integrates all levels, including national goals and funding, mass awareness campaigns, continuing medical education, establishing a standard of practice for screening, community-level peer-education, and individual tracking and counseling of patients.

Findings from this study are especially relevant as national plans for implementing cancer screening services are being rolled out in the country [16] and as the latest data suggests that current screening rates in the country are less than 30% [3]. The participants' insights highlight the need for future research to gather perspectives from other key stakeholders including government officials, clinical service administrators, laboratory personnel, communities, and women. In the proposed government program, physicians (noted as medical officers) carry the primary responsibility of delivering the cervical cancer screening services at the public health care setting. Of important note are the roles of nurses who support physicians in providing cervical cancer screening in the setting and CHWs who act as liaisons to the communities to recruit eligible women for cervical cancer screening. Incorporating their perspectives in future research studies will be crucial for understanding how best to implement cervical cancer screening for populations benefit. One such study examined various perspectives on cervical cancer prevention and treatment in Uttar Pradesh, India in 2005, and similarly concluded that

an integrated approach would be required for cervical cancer management [20]. However, there remains a dearth of information for how to implement cancer prevention programs on the national level.

We acknowledge the limitations of this study in that we interviewed physicians from one geographical location (Mysore, India) which may have limited generalizability to other low-resource communities in India and beyond. However, the organization of the primary health-care setting including the structure of physicians and community health workers is consistently similar across the districts in India. Therefore, we believe that the findings from our study in Mysore, India, can have important implications for implementation of cancer screening services across primary care settings in India. We only interviewed physicians but acknowledge that perspectives from supporting staff such as nurses and CHWs in primary care settings are also important. Additionally, we recruited participants using convenience sampling and a snowball system, which, while suitable for generating a preliminary understanding of provider perspectives, may represent limited viewpoints. Future research aimed at gathering perspectives from all key stakeholders involved in specific implementation efforts, can be provide specific information regarding the intervention points. Finally, our study gathered perspectives from physicians representing different specialties. Future work may look at individual medical specialties for even deeper understanding of their perspectives.

The insights from the providers lead to important and innovative recommendations for the implementation of a cervical cancer screening program. Fig 2 highlights several key

**Key recommendations:**

- Educational interventions that address both the low health literacy and reduction of stigma around gynecologic health and cancer

- Utilize community-level approaches to promote a culture of prevention including but not limited to community health workers

- Importance of non-pelvic exam screening interventions to overcome personal barriers to gynecologic health

- Assess and address healthcare organization's capacity in terms of providers and resources for effective delivery of cancer screening services

- Promote awareness among physicians about cancer prevention and provide trainings for implementing cancer screening service

**Fig 2. Key recommendations for implementation of cervical cancer prevention programs based upon interviews with thirty physicians in Mysore, India.**

recommendations for how to achieve national implementation of cancer screening programs in primary care settings. Educational interventions should utilize the existing network of CHWs and focus on improving health literacy and reducing stigma in order to promote gynecologic health. There is limited research in the context of LMICs regarding interventions for reducing stigma and improving health literacy surrounding cancer screening [21]. Interventions using mobile health technology show acceptability among CHWs but do not address social and cultural barriers [22]. Expanding educational interventions to target socio-cultural barriers should be informed by socio-behavioral theories and require efforts informed by community-based participatory research [23].

Beyond education, testing strategies that do not require pelvic exams (i.e. self-sampling) could be acceptable screening options. For example, molecular tests that can screen for presence of high-risk HPV DNA are highly accurate [24] and offer the distinct advantage of self-sampling [24–27]. Review articles have found improved participation among women who do not routinely attend screening programs and that self-sampling may be particularly suited to indigenous communities and populations typically under screened in lower socioeconomic groups [28–30]. Studies examining the feasibility of self-sampling from India show promise for further research [31,32].

Efforts must also target delivery of clinical services by assessing healthcare organizations' capacity and readiness, which can inform the development of context-specific strategies to implement screening programs. Tools such as the Improving Data for Decision Making in Global Cervical Cancer can be used to conduct organizational assessments prior to implementation and inform the need for additional resources [33]. For physicians working in the public sectors, continuing medical education can improve knowledge and skills for implementing cancer-screening services in primary care settings. Initiatives such as Project ECHO at the Indian National Institute for Cancer Prevention and Research have demonstrated effectiveness in improving knowledge among healthcare providers [34,35]. Investing in further research to evaluate these trainings can help identify and develop strategies to address gaps that can promote the implementation of cancer screening services.

## Conclusion

Findings from the qualitative inquiry with physicians from one district in India suggest the need for a multi-level, integrated strategy for implementing cervical cancer screening programs in their communities. Based on these findings, we recommend future research efforts directed towards developing and testing culturally-tailored educational strategies, promoting screening tests that do not require pelvic examinations, utilizing community health workers to promote uptake in communities, assessing and addressing healthcare organizations' capacity for delivery of services, and providing physician training and continuing education in cancer prevention. We believe that collectively, these recommendations can contribute to achieving the population level outcomes of reduced mortality due to cancer by promoting the effective implementation of cancer screening programs.

## Acknowledgments

We sincerely thank the study participants for their insights and time.

## Author Contributions

**Conceptualization:** Prajakta Adsul, Sasha Herbst de Cortina, Rashmi Pramathesh, Poornima Jayakrishna, Vijaya Srinivas, Purnima Madhivanan.

**Data curation:** Prajakta Adsul, Rashmi Pramathesh, Vijaya Srinivas.

**Formal analysis:** Sasha Herbst de Cortina.

**Funding acquisition:** Prajakta Adsul, Vijaya Srinivas, Purnima Madhivanan.

**Investigation:** Purnima Madhivanan.

**Methodology:** Poornima Jayakrishna, Suzanne Tanya Nethan, Kavitha Dhanasekaran, Roopa Hariprasad, Purnima Madhivanan.

**Project administration:** Poornima Jayakrishna, Vijaya Srinivas.

**Resources:** Rashmi Pramathesh, Purnima Madhivanan.

**Supervision:** Poornima Jayakrishna.

**Writing – original draft:** Prajakta Adsul, Sasha Herbst de Cortina, Rashmi Pramathesh, Poornima Jayakrishna, Vijaya Srinivas, Suzanne Tanya Nethan, Kavitha Dhanasekaran, Roopa Hariprasad, Purnima Madhivanan.

**Writing – review & editing:** Prajakta Adsul, Sasha Herbst de Cortina, Rashmi Pramathesh, Poornima Jayakrishna, Vijaya Srinivas, Suzanne Tanya Nethan, Kavitha Dhanasekaran, Roopa Hariprasad, Purnima Madhivanan.

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
