## [Decision Letter · Decision Letter 0]

18 Feb 2022

PGPH-D-21-00418

Asking physicians how best to implement cervical cancer prevention services in India: a qualitative study from Mysore

Dear Dr. Adsul,

Thank you for submitting your manuscript to PLOS Global Public Health. After careful consideration, we feel that it has merit but does not fully meet PLOS Global Public Health’s publication criteria as it currently stands. Therefore, we invite you to submit a revised version of the manuscript that addresses the points raised during the review process.

We look forward to receiving your revised manuscript.

Kind regards,

Veena Sriram

Academic Editor

Journal Requirements:

1. We see that your study includes live participants, but you have not included an Ethics Statement. Please update your manuscript file to include an Ethics Statement subsection to your Materials and Methods section. It should include:

i) The full name(s) of the Institutional Review Board(s) or Ethics Committee(s)

2. Please provide separate figure files in .tif or .eps format only.  Please ensure that all files are under our size limit of 20MB.  

For more information about how to convert your figure files please see our guidelines: Once you've converted your files to .tif or .eps, please also make sure that your figures meet our format requirements

Additional Editor Comments (if provided):

Thank you for your submission, and our apologies for the delay in sharing a decision. As you will see from the reviews, there was a split decision between the reviewers between Major and Minor Revisions. Based on R1's comments, we have labelled this as a Major Revision, and invite you to submit a revision responding to Reviewer comments. Thanks again for your patience!

Reviewers' comments:

Reviewer's Responses to Questions

**Comments to the Author**

1. Does this manuscript meet PLOS Global Public Health’s publication criteria? Is the manuscript technically sound, and do the data support the conclusions? The manuscript must describe methodologically and ethically rigorous research with conclusions that are appropriately drawn based on the data presented.

Reviewer #1: Partly

Reviewer #2: Partly

2. Has the statistical analysis been performed appropriately and rigorously?

Reviewer #1: N/A

Reviewer #2: Yes

3. Have the authors made all data underlying the findings in their manuscript fully available (please refer to the Data Availability Statement at the start of the manuscript PDF file)?

Reviewer #1: Yes

Reviewer #2: Yes

4. Is the manuscript presented in an intelligible fashion and written in standard English?

Reviewer #1: Yes

Reviewer #2: No

5. Review Comments to the Author

Reviewer #1: There is currently limited research in the context of the Indian healthcare system and perceptions of physicians regarding cancer screening services. Most studies to date are quantitative and not qualitative therefore the aim of this study was to use qualitative interviews to understand the perspectives of a group of physicians (PCPs, OBGYN, oncologists, pathologists) involved in cervical cancer related preventative services using the multi-level influences of cancer care continuum model. This is an important and relevant topic. Overall the paper needs re-organization and re-thinking of the emerging themes to interpret the quotations accurately and connect better to the conclusions.

In the methods section, would shift the last few sentences "according to the India's Fourth National....and "to our knowledge" to the Background. A few clarifying questions - are there any OB/GYNs in the public sector in Mysore and if so, why were they excluded? Is it possible to provide a diagram of the snowball sampling to understand from whom the referrals were made (within the public vs. private sectors). Why were pathologists interviewed? They would review samples but not performing any screening. Were the primary care doctors in the public sector specialists that completed residency or medical officers that completed an MBBS (medical school plus intern year)?

For the results, I would integrate important quotations into the results and place the larger results table in the Appendix. In general for the quotations, please provide the job title and sector from where the quotations come from. For Theme 5 -- it would be a very concerning major gap if gynecologists reported requiring more training or didn't have a clear sense of when cervical care screenings should be done (as prevention) than pathologists or other physicians.

A major area of revision for this paper includes: The themes are not always consistent with the quotations and would consider re-wording and re-framing a few themes. It seems like quote 3/quote 4 might have to do with cultural norms, gender inequality and stigma rather than solved by promoting health literacy. Quote 5 seems to describe difficulties with the economic repercussions of trying to access care (missing work). When going into this into the results, would consider splitting the themes and re-organizing since the text under Theme 1 jumps from a lot of different results that are not directly related to Theme 1. Quote 7 seems to deal with community influence and lack of trust in medical systems more than stigma (possibly more lack of trust in the medical system and community perceptions of what happens at the hospital rather than judgment towards the patient for having cancer). Again, quote 8 seems to be a quote more on not valuing elderly women (not valuing health care services for women and the financial burden of being diagnosed with cancer) rather than stigma towards cervical cancer.

Finally, It’s not clear how the multilevel influences on the cancer care continuum was used to organize themes --- would do a better job of tying this framework to the results and conclusions and Figure 1.

Reviewer #2: The concept is good however some points needs to be explained in more detailed manner. This

Is with regard to the process of conducting the study. Clarity is to be mentioned

regarding the setting, involving opinion of community workers etc. Application and generalusability

to be implemented also needs to be elaborated.

Revision of manuscript is needed before consideration for acceptance

6. PLOS authors have the option to publish the peer review history of their article (what does this mean?). If published, this will include your full peer review and any attached files.

**Do you want your identity to be public for this peer review?** For information about this choice, including consent withdrawal, please see our Privacy Policy.

Reviewer #1: No

Reviewer #2: No

---

## [Decision Letter · Decision Letter 1]

11 May 2022

Asking physicians how best to implement cervical cancer prevention services in India: a qualitative study from Mysore

PGPH-D-21-00418R1

Dear Dr. Adsul,

We are pleased to inform you that your manuscript 'Asking physicians how best to implement cervical cancer prevention services in India: a qualitative study from Mysore' has been provisionally accepted for publication in PLOS Global Public Health.

Best regards,

Veena Sriram

Academic Editor

Many thanks for your patience! We are pleased to accept this paper, and wanted to share a few minor suggestions below for your consideration for the final version of the manuscript.

Page 3, line 65

like India – consider making this, such as India.

Page 3, line 81

5 years – consider writing as five years

Page 4-5, lines 129 – 131

No studies conducted in India on this topic? If so, pls clarify this.

Page 6, lines 142 – 144

Our goal was to capture both aspects of the screening process, delivery of the tests as well as the results, and hence we included pathologists in our study. => pls consider writing as: “Our goal was to capture both aspects of the screening process – delivery of tests and results – and therefore, we included pathologists in our sample.”

Page 7, line 166

Pls consider writing as “PA selected 20% of the transcripts at random to review…”

Page 7

Methods para here seems to go between first person and third person; pls consider turning into third person for consistency.

Page 21, line 297

Additional support rather than addition support?

Reviewer Comments (if any, and for reference):

Reviewer's Responses to Questions

**Comments to the Author**

1. If the authors have adequately addressed your comments raised in a previous round of review and you feel that this manuscript is now acceptable for publication, you may indicate that here to bypass the “Comments to the Author” section, enter your conflict of interest statement in the “Confidential to Editor” section, and submit your "Accept" recommendation.

Reviewer #1: All comments have been addressed

2. Does this manuscript meet PLOS Global Public Health’s publication criteria? Is the manuscript technically sound, and do the data support the conclusions? The manuscript must describe methodologically and ethically rigorous research with conclusions that are appropriately drawn based on the data presented.

Reviewer #1: Yes

3. Has the statistical analysis been performed appropriately and rigorously?

Reviewer #1: Yes

4. Have the authors made all data underlying the findings in their manuscript fully available (please refer to the Data Availability Statement at the start of the manuscript PDF file)?

Reviewer #1: Yes

5. Is the manuscript presented in an intelligible fashion and written in standard English?

Reviewer #1: Yes

6. Review Comments to the Author

Reviewer #1: All original concerns were adequately addressed and I believe this to be an important contribution to cervical cancer screening and prevention research.

7. PLOS authors have the option to publish the peer review history of their article (what does this mean?). If published, this will include your full peer review and any attached files.

**Do you want your identity to be public for this peer review?** For information about this choice, including consent withdrawal, please see our Privacy Policy.

Reviewer #1: No
